# Cryopreservation Competence of Chicken Oocytes as a Model of Endangered Wild Birds: Effects of Storage Time and Temperature on the Ovarian Follicle Survival

**DOI:** 10.3390/ani12111434

**Published:** 2022-06-02

**Authors:** Mayako Fujihara, Jun-ichi Shiraishi, Manabu Onuma, Yoshiyuki Ohta, Miho Inoue-Murayama

**Affiliations:** 1Wildlife Research Center, Kyoto University, Kyoto 606-8203, Japan; murayama.miho.5n@kyoto-u.ac.jp; 2Graduate School of Applied Biochemistry, Nippon Veterinary and Life Science University, Tokyo 180-8602, Japan; jshira@nvlu.ac.jp (J.-i.S.); ohta-y@nvlu.ac.jp (Y.O.); 3National Institute for Environmental Studies, Tsukuba 305-8506, Japan; monuma@nies.go.jp

**Keywords:** ovary, oocyte, cryopreservation, birds, chicken

## Abstract

**Simple Summary:**

Many avian species are classified as endangered, and the development of the gamete preservation technique is essential to preserve their genetic diversity. Since cryopreservation and transplantation of ovarian tissues are currently the only methods of female fertility preservation in birds after birth, transporting the ovary to the laboratory in good condition is key for successful female fertility preservation. Here, we report using chickens as a model of wild birds for which storage at low temperature (4 °C), but not room temperature, can maintain the ovary, ensuring oocyte survival, morphology and protein expression of germ cell markers for at least 3 days. Furthermore, a vitrification study confirmed that oocytes after ovary storage at a low temperature can be preserved by ovarian tissue cryopreservation. On the other hand, ovaries stored at room temperature decreased oocyte survival, while increasing gene expression of apoptosis and oxidation stress markers. Our study not only verifies the fact that the storage of hen ovaries at low temperatures protects oocytes, which can be preserved by vitrification, but it also contributes to understanding the effect of ovary storage on the integrity and cryotolerance of immature follicles within the hen ovary.

**Abstract:**

For the conservation of endangered avian species, developing gamete preservation technologies is essential. However, studies in oocytes have not been widely conducted. In this study, assuming that the ovaries are transported to a research facility after death, we investigated the effect of ovary storage on oocytes for the purpose of cryopreserving avian female gametes by using a chicken as a model of endangered avian species. After excision, the ovaries were stored at either a low temperature (4 °C) or room temperature for 1–3 days. Ovarian follicles stored under different conditions for each period were examined by neutral red staining, histology, and gene and protein expression analysis. In addition, the pH of the storage medium after preserving the ovaries was measured. Then, ovarian tissues were vitrified to determine the cryopreservation competence. Storing the ovarian tissues at 4 °C kept the follicles viable and morphologically normal for 3 days with slow decline. In contrast, although different storage temperature did not influence follicle viability and morphology after only 1 day of storage, ovarian tissues stored at room temperature rapidly declined in structurally normal follicles, and viable follicles were rarely seen after 3 days of storage. Gene and protein expression analysis showed that apoptosis had already started on the first day, as shown by the higher expression of CASP9 under room temperature conditions. Furthermore, high expression of *SOD1* and a rapid decline of pH in the storage medium under room temperature storage suggested the influence of oxidative stress associated with low pH in this condition on the follicle survivability in hen ovarian tissues. Our cryopreservation study also showed that ovarian tissues stored at 4 °C could recover after cryopreservation even after 3 days of storage. The described storage conditions and cryopreservation methods, which preserve chicken follicle survival, will lay the foundation of ovarian tissue preservation to preserve the fertility of wild female birds.

## 1. Introduction

The International Union for Conservation of Nature’s (IUCN’s) Red List of Threatened Species warns that as many as 13% of birds are listed as endangered [1]. In this respect, gamete preservation plays an important role in the conservation of genetic diversity and creating reproductive opportunities for more individuals of endangered species. However, research in avian gamete preservation has gained less attention than that in mammals. This is essential for the preservation of species, especially of endangered birds, such as the Okinawa rail (*Gallirallus okinawae*), which is an endangered endemic species in the northern part of Okinawa Island, Japan [2]. For the development of genetic preservation technology for avian species, the primordial germ cells (PGCs), which are progenitor germ cells in the embryonic stage, are widely studied for the purpose of conservation of avian genetic resources [3]. While the preservation of PGCs is a promising technique in combination with chimera generation, the feasibility of this strategy for genetic conservation is limited, especially for conservation purposes, because it can be useful only in specific cases, when eggs that do not hatch are obtained. Alternatively, research on semen freezing has been actively carried out in domestic avian species for poultry production. More recently, semen cryopreservation has started to be studied in non-domestic birds, such as endangered cranes [4]. Meanwhile, the cryopreservation of female germplasm has been challenging in females due to the characteristics of avian ovaries and eggs with a large amount of yolk, which prevent the application of the cryopreservation techniques that have been used for mammals.

The chicken has a single ovary with different-sized follicles, which enclose primary oocytes, including a primordial follicle, primary follicles, and a pre-hierarchical follicle stage. While cryopreservation of ovarian tissue has been studied in mammals as a potential method to preserve female fertility, including in women threatened by cancer therapies [5], it has been also attempted in hens [6,7] and Japanese quail [8,9]. Moreover, some studies have revealed that ovarian tissue cryopreservation can be of use for transplantation, resulting in successful birth, for genetic preservation purposes [8,10,11]. Therefore, ovarian tissue cryopreservation is a reliable technique for the preservation of female genetic resources of endangered avian species. Since the combination techniques of cryopreservation and transplantation of ovarian tissues are currently the only methods of female fertility preservation in birds after birth, it is essential to cryopreserve the ovarian tissue in good condition.

Traffic accidents are a common threat for wild birds, including endangered species such as Okinawa rail, and there is a need to take the opportunity of obtaining the reproductive tract and germplasm in situ to use for conservation purposes [12]. In general, habitats of wild endangered birds are far from the laboratory. To recover their germplasms to use for cryopreservation, ovaries need to be stored in a good condition for a long period of time until ovarian tissues are cryopreserved. In mammals, the impact of different storage times and temperatures on the quality of oocytes is different among species. For instance, the maturation rates of oocyte nuclei decrease with the increase in ovary storage time in pigs [13], but do not change in horses [14]. The storage temperature of the ovary for transportation affected oocyte meiotic competence in vitro in cats [15], dogs [16] and pigs [13] and changed the cleavage and blastocyst formation rates in bovines [17]. However, little information is available regarding the effects of ovary storage conditions on ovarian tissue cryopreservation. And moreover, to our knowledge, no reports have been published on the influence of ovary storage conditions in avian species.

The overall goal of the present study was to understand appropriate shipping/storage conditions for avian ovaries that can sustain female gamete preservation in birds. By using ovaries of laying hens (*Gallus gallus domesticus*) as a model animal of endangered avian species, we investigate different storage conditions for the purpose of cryopreserving avian ovarian tissues. Our specific objectives were to determine the influences of (i) storage temperature (low (4 °C) vs. room temperature) and (ii) the storage period (1, 2 and 3 days) on the viability and morphology of follicles within hens’ ovaries as well as gene/protein expression. Cryopreservation experiments were also carried out to evaluate if the storage methods optimally maintain immature follicles within the chicken ovarian tissues.

## 2. Methods

### 2.1. Chemicals

All chemicals were purchased from Sigma-Aldrich (St. Louis, MO, USA) unless otherwise indicated.

### 2.2. Collection of Chicken Ovarian Tissues

The ovaries of laying hens of White Leghorn breed (*Gallus gallus domesticus*) (1 year old; *n* = 30) maintained at the Nippon Veterinary and Life Science University (Tokyo, Japan) were used. Ovaries were removed immediately after sacrifice by decapitation, rinsed with PBS, and immersed in storage medium of L-15 containing 10 mM HEPES, 100 μg/mL streptomycin sulphate (FUJIFILM Wako Pure Chemical Corporation, Osaka, Japan), 100 μg/mL penicillin G sodium (FUJIFILM Wako Pure Chemical Corporation), and 0.5 mM Ascorbic acid (pH 7.1) [18,19]. Ovaries in the storage medium were then transported to the laboratory either at 4 °C or room temperature and stored under the same conditions as the shipping temperature for 1–3 days until use. The pH of the storage medium was measured at the end of the storage in each ovary.

Ovaries were rinsed and then dissected in working medium of Hanks’ MEM (Gibco, Thermo Fisher Scientific, Waltham, MA, USA) supplemented with 15 mM Hepes, 1% BSA, 2 mM glutamax (Thermo Fisher Scientific), 100 μg/mL streptomycin sulphate (FUJIFILM Wako Pure Chemical Corporation), and 100 μg/mL penicillin G sodium (FUJIFILM Wako Pure Chemical Corporation) [18]. Follicles larger than 2 mm were mechanically removed from the ovaries (Figure 1A) and tissues were cut into equal pieces (5 mm^2^). Because the follicles are not evenly distributed in the ovaries, at least two pieces of ovarian tissues from each donor stored under various conditions were utilised for each experiment. In some of the ovaries stored at 4 °C for different periods, the remaining ovarian tissues were vitrified.

### 2.3. Neutral Red (NR) Staining to Assess Viable Ovarian Follicles in Fresh and Vitrified–Warmed Tissues

At least two pieces of fresh or vitrified–warmed ovarian tissues were rinsed in MEM working medium and then incubated in 0.5 mL of enzyme solution consisting of 1.5 mg/mL collagenase and trypsin/ETDA in MEM working medium in the 5% CO_2_ incubator for 2.5 h at 38.5 °C with gentle agitation. The enzymatic digestion of ovarian tissues was stopped by addition of the KAv-1 medium, MEM-based medium containing 5% fatal bovine serum and 5% chick serum [20]; then, ovarian tissues were incubated with Kav-1 medium supplemented with 33 μg/mL of NR solution (2-amino-3 methyl-7-dimethyl-aminophenazoniumchloride) in the CO_2_ incubator for 2 h at 38.5 °C. Follicles consisting of an oocyte surrounded by granulosa cells were easily distinguished from stromal cells within the ovarian tissues after incubation with enzyme solution. The number of red-coloured follicles stained with NR (viable) and uncoloured (dead) follicles were counted in each ovarian tissues under a stereomicroscope (Nikon SMZ800N, Nikon, Tokyo, Japan). Follicles were only considered to be viable when both the oocyte and more than 75% of the surrounding granulosa cells stained positive for NR (Figure 1B).

### 2.4. Histological Assessment and Classification of Follicular Structure

Histological assessment and classification of follicle structure were conducted as described previously with slight modification [18,21]. Briefly, at least three pieces of ovarian tissue in each group were fixed with Bouin’s solution (FUJIFILM Wako Pure Chemical Corporation), kept at 4 °C overnight, dehydrated in a series of ethanol solutions (70–100%), and embedded in paraffin. Serial sections (5 μm thick) of ovarian tissues were prepared and stained with haematoxylin and eosin (both Muto Pure Chemicals CO. Tokyo, Japan). To avoid double counting, three sections, each at least 20 µm apart, were assessed by light microscopy (Nikon ECLIPE E600, Nikon).

Follicles within the ovarian tissues were classified as primordial (<80 um), primary (0.08–1 mm), or pre-hierarchical (>1 mm) [8]. All the follicles were further characterized as follows: ‘normal’, when the layer of granulosa cells is attached to the spherical oocyte surrounding it and the homogenous ooplasm contains a tiny granulated nucleus, or ‘abnormal’, wherein aggregation and shrinkage of nuclear chromatin and wrinkling of the nuclear membrane were regarded as signs of atresia [6]. The proportion of morphologically normal follicles per section was calculated by dividing the number of normal follicles by the total number of assessed follicles.

### 2.5. Gene Expression Analysis by Quantitative Reverse Transcription Polymerase Chain Reaction (qPCR)

The expression of caspase 3 (*CASP3*) and 9 (*CASP9*) as well as superoxide dismutase 1 (*SOD1*) was determined by qPCR using ovaries under different storage conditions. Two to three tissues recovered from ovaries stored at different temperatures for 1–3 days were soaked in RNAlater (Invitrogen, Carlsbad, CA, USA) and stored at −80 °C until RNA extraction. After thawing, RNA was extracted by using the RNeasy Mini kit (QIAGEN, Hilden, Germany) and reverse transcription was performed with a Prime Script RT-PCR kit (Invitrogen), both according to the manufacturer’s instructions. For each sample, qPCR was completed in triplicate using SYBR Pemix Ex Taq II (TaKaRa, Shiga, Japan) and a Thermal Cycler Dice Real-Time System TP800 (TaKaRa). PCR amplification was conducted with 40 cycles (95 °C for 5 s and 60 °C for 30 s). Amplification and melting curves of each PCR product were checked to verify efficiencies and the targeted amplicon. The CT value of each gene was normalized against glyceraldehyde-3-phosphate dehydrogenase (GAPDH, housekeeping gene). The qPCR data expressed as the relative mRNA expression and fold change over ovaries stored at room temperature for 1 day were analysed with the machine’s software (Thermal Cycler Dice Real Time System TP800 Software, ver. 5.11B, TaKaRa). The primers for specific transcripts were designed using the Primer3 software, with all the sequences listed in Table 1.

### 2.6. Protein Expression Analysis by Western Blotting

Ovarian tissues stored at different temperatures and for different periods were extracted in EzRIPA Lysis kit (ATTO, Tokyo, Japan) with phosphatase and protease inhibitors, and the samples were separated on 10 to 20% e-PAGEL (ATTO) by SDS-PAGE. Separated proteins were electroblotted onto polyvinylidene difluoride membranes (ATTO). Precision Plus Protein™ Western C Standards (Bio-Rad, Hercules, CA, USA) were used as the molecular weight standard. The blots were blocked with Ez Block Chemi (ATTO) for 1 h (~22 °C) and incubated overnight (4 °C) with the respective primary antibodies: rabbit anti-Caspase 9 (Enzo Life Science, Farmingdale, NY, USA, 1:1000, 46 kDa), rat anti-Chicken-Vasa-Homologue (CVH, generously provided by Prof. Nobuhiko Yamauchi, Kyushu University, 1:3000, 75 kDa) [22,23] or mouse anti-GAPDH (Proteintech, Rosemont, IL, USA, 1:5000 in TBS-T, 36 kDa). Anti-rabbit, anti-rat or anti-mouse IgG antibodies (Cell Signaling Technology, Danvers, MA, USA 1:2000) conjugated with horseradish peroxidase (HRP) were used as the secondary antibodies, and blots were incubated for 1 h (~22 °C). Immunoreactivity was visualized with ECL Prime Western Blotting Detection Reagent (GE Healthcare, Chicago, IL, USA).

### 2.7. Vitrification and Warming Procedure

The ovarian tissue vitrification was carried out using a needle immersion vitrification protocol described previously [9] with slight modification. Briefly, ovarian tissues were threaded onto a 27 G needle (5–6 pieces per needle; Terumo Corporation, Tokyo, Japan) with space between each piece. The needles that pierced the tissue were rinsed in KAv-1 medium at first and immersed in equilibration solution consisting of 7.5% dimethyl sulfoxide (DMSO) and 7.5% ethylene glycol (EG) in KAv-1 medium for 10 min at 4 °C. The needles were subsequently moved into vitrification solution consisting of 15% DMSO, 15% EG, and 0.5 M sucrose in KAv-1 for 10 min at 4 °C. Then, after excess solution was absorbed quickly with a Kimwipe, the needles were plunged into liquid nitrogen directly, inserted into a cryovial (Watson, Tokyo, Japan) containing liquid nitrogen, and stored for at least 1 week.

For warming, needles holding tissue pieces were taken out of the cryovial in liquid nitrogen and immediately transferred into a washing gradient solution. The vitrified tissues held by the needles were washed in gradient solution (KAv-1 supplemented with 1, 0.5, 0.25, and 0 M sucrose) for 5 min at each step at 37 °C. The ovarian tissues were then removed from the needles for further assessment.

### 2.8. Experimental Design

#### 2.8.1. Study 1: Effect of Different Storage Conditions on Follicle Maintenance in Chicken Ovaries

The ovary from each animal was randomly allocated to one of two storage temperatures, either low (4 °C) or room temperature. Then, ovaries and storage medium at each storage temperature were examined at different storage periods (1, 2 and 3 days). Hence, we divided 30 hens into five replicates of six different storage conditions (three storage periods of room vs. low temperature). The effect of different storage temperatures and times on ovarian follicles were evaluated by pH analysis in storage medium (*n* = 5 per condition), then follicle normality by NR staining and histology (*n* = 4 per condition in each analysis, at least 2 pieces/hen/condition) in ovarian tissues that were separately stored under different conditions. Gene and protein expressions were then evaluated by qPCR and Western blotting (*n* = 3 per condition in each analysis, 2–3 pieces/hen/condition).

#### 2.8.2. Study 2: Examine the Freezing Tolerance of Ovarian Follicles Stored for Different Storage Periods

Based on the results of Study 1, ovaries stored at low temperatures (4 °C) were vitrified for different storage periods (1, 2 and 3 days) to investigate the effect of storage time by vitrification on freezing tolerance of ovarian follicle. After warming, vitrified–warmed tissues were examined with NR staining and ovarian follicle histology (*n* = 4 per storage period in each analysis, at least 2 pieces/hen/storage period).

### 2.9. Statistical Analysis

Each value shown is the mean with the standard error of mean (SEM). The Shapiro–Wilk test was used to determine if the dataset was normal. Additionally, the Bartlett test was used to evaluate the homogeneity of variance. Analysis of variance (ANOVA) followed by a Turkey’s multiple comparison test were applied to compare follicle viability and morphology, pH of storage medium and relative expression level of mRNA among different storage conditions or before/after vitrification. Differences were considered significant at a level of *p* < 0.05 (GraphPad PRISM 9, GraphPad Software, La Jolla, CA, USA).

## 3. Results

### 3.1. Changes in pH of Storage Medium and Follicle Viability in Chicken Ovarian Tissues under Different Storage Conditions

A starting point of our analysis was to investigate the effect of different storage temperatures (low vs. room temperature) and different storage periods (1 to 3 days) on the pH of the medium in which the hen ovary was stored. While the pH of the storage medium did not change much after 1 day of storage in the low-temperature conditions (from pH 7.1 to pH 7.0), the pH value decreased greatly at room temperature (pH 6.7) compared to before storage and to that at the low temperature at 1 day of storage (Figure 1C). At the low temperature, the pH of the storage medium was slightly decreased in a time-dependent manner, but the changes were maintained as minimal for 3 days of storage from pH 7.0 to pH 6.9. On the other hand, at room temperature, although the pH of the storage medium was maintained from day 1 to day 2 of storage at around pH 6.7, it dropped rapidly to pH 6.5 after 3 days of storage.

The viability of ovarian follicles stored under different conditions for each period was evaluated by NR staining (Figure 1B). The viability of the follicles within the stored ovaries was also decreased in a time- and temperature-dependent manner. At 1 day after storage, both temperature groups had follicles with positive NR staining, which were considered to be viable. The difference in viability among different storage temperatures was small, and ovaries stored at a low temperature had slightly more viable follicles (76.1 ± 0.8%) compared to those at room temperature (66.5 ± 1.1%, Figure 2A). However, the difference in viability in ovarian follicles was increased as storage time increased. In the low-temperature group, although the viability of ovarian follicles was gradually decreased, 52.6 ± 1.3% of the follicles within the ovarian tissues still survived after 3 days of storage (Figure 2A). On the other hand, in the room-temperature group, the viability of ovarian follicles was already decreased to 42.0 ± 1.8% at 2 days of storage, and surviving follicles were rarely seen (3.5 ± 1.6%) after 3 days of storage (Figure 2A).

Histological examination revealed that ovarian tissues examined for the current study were mostly (>98%) primordial and primary follicles and more than 80% of them were intact on the first day of storage in both temperature groups (Figure 2B,C). However, at 2 days of storage, while the low-temperature condition maintained the proportion of intact follicles (81.2 ± 1.5%), ovarian tissues in the room-temperature condition tended to decrease the proportion of intact follicles to 48.5 ± 2.0% (Figure 2B,C). The difference in follicle maintenance became significantly different (*p* = 0.01) after 3 days of storage. Although ovarian tissues stored under the low-temperature conditions still kept 60.0 ± 2.6% of intact follicles after 3 days of storage, intact follicles were rarely seen (3.7 ± 1.1%) in the room-temperature conditions at the same storage period (Figure 2B,C).

### 3.2. Influence of Different Storage Conditions on Gene and Protein Expressions in Chicken Ovarian Tissues

Gene expression analysis showed that expression level of *CASP9,* an apoptosis marker gene, tended to be higher in the room-temperature group compared to the low-temperature group at the first day of storage (*p* = 0.56, Figure 3). In the low-temperature group, expression of *CASP9* was maintained as low for 2 days of storage but became up-regulated at 3 days. A different apoptosis marker gene, *CASP3*, did not differ among storage condition and time (Figure 3). The analysis of *SOD1* expression, a marker of oxidation stress, indicated a trend of higher expression levels in the room-temperature group than in the low-temperature group (Figure 3). In both conditions, *SOD1* expression tended to increase as storage time increased (Figure 3).

Western blotting analysis revealed higher expression of CASP9 in the room-temperature group than in the cold group, especially on the first day of storage (Figure 4). On the other hand, CASP9 protein expression in the low-temperature group remained low for 3 days of storage (Figure 4). The expression of the germ cell marker, Chicken Vasa Homologue (CVH), was also determined in the ovarian tissues in both group for each storage time (Figure 4).

### 3.3. Influence of Storage Time on Cryopreservation of Chicken Ovarian Tissues

To understand how ovary preservation might affect the freezing tolerance of ovarian follicles, follicular viability and morphological maintenance were evaluated after ovarian tissue vitrification by NR staining and histological analysis. Based on the results of Study 1, ovaries stored in low-temperature conditions for different storage times were utilized for vitrification (*n* = 4/storage time).

NR staining revealed that all of the vitrified ovarian tissues had follicles with positive NR staining, which were considered to be viable (Figure 5A). The proportion of viable follicles within ovarian tissues significantly decreased after vitrification at each preservation time (*p* < 0.005, Figure 5A). Similar to the ovarian tissues before vitrification, the percentage of viable follicles tended to decrease in vitrified tissues as preservation time was increased, but approximately 20% of follicles still survived after vitrification in the ovarian tissues preserved for 3 days prior to vitrification. (*p* = 0.12, Day 1, 38.1 ± 2.1%; Day 2, 32.5 ± 2.2; Day 3, 19.6 ± 2.0%; Figure 5A).

Histological observations of vitrified tissues showed the presence of structurally intact follicles with visually healthy nuclei in all the ovarian tissues regardless of the preservation time (Figure 5B,C). Specifically, ovarian tissues preserved for 1 day retained 68.9 ± 1.5% of intact follicles after vitrification (Figure 5B). This value did not significantly change as preservation time increased, but it tended to decrease as preservation time increased. Nevertheless, histological observation clearly showed that the ovarian tissues preserved over 2 days still hold the morphologically normal follicles (Day 2, 57.1 ± 2.7%; Day 3, 46.9 ± 2.2%. Figure 5B).

## 4. Discussion

This work demonstrates the relationship between oocyte viability and storage time and temperature in avian ovaries for the first time. Cryopreservation of ovarian tissue is a promising approach for female fertility preservation, and successful transportation of the ovary is a key factor for effective cryopreservation of avian female gonads as a means of genetic preservation. Using a chicken as a model for wild avians, this study was the first to evaluate the storage condition of ovaries for the purpose of ovarian tissue cryopreservation based on viability, morphology and gene/protein expressions of follicles and its cryotolerance in this species. Specifically, the present study demonstrates that a low (4 °C) temperature can successfully maintain the viability and morphology of primordial and primary follicles within hen’s ovaries for 3 days. Furthermore, it was clear that follicles in the stored ovaries can survive after vitrification.

Trends in follicle viability affected by different storage conditions were similar between NR staining and histological analysis. In the present study, ovarian tissues stored at room temperature had lower cell viability and fewer morphologically normal follicles compared with their counterparts stored at a low temperature throughout the ovary storage. Furthermore, while the room-temperature condition did not retain follicle survival for 3 days of storage, low-temperature conditions maintained follicle survival and, moreover, cryotolerance determined by vitrification assay. A study in cats showed that ovary storage at 4 °C does not affect the meiotic competence of oocytes in vitro until 24 h of storage, but it decreases dramatically after 48 h of storage [15]. Here, our study demonstrated in hen ovaries that ovary storage at the same temperature surely decreased the follicle viability before and after vitrification in a time-dependent manner, but the decline did not occur rapidly compared to storage at room temperature. Moreover, especially in the room-temperature condition, we observed a decrease in the pH of the medium throughout ovary storage. In pig studies, decreasing pH was observed in the follicular fluid during ovary storage, with a more pronounced tendency at higher temperatures [24]. Coincidentally, follicle viability evaluated by NR staining and histological analysis decreased as pH decreased, therefore suggesting that oxidation stress caused the damage to ovarian follicles.

For the first time, the expression of some apoptosis-related genes and proteins in stored hen ovaries was measured. In general, caspase (cysteine-dependent aspartate-directed proteases) has a determinative role and activation in apoptosis. Activation of the caspase cascade results in cleavage and disintegration of substrate proteins and subsequent removal of the degenerating cell [25]. Our findings indicated that *CASP9* was strongly expressed in the ovarian tissues stored at room temperature compared to the low-temperature group on the first day of ovary storage. While the highest expression of *CASP9* was seen on the first day of storage in the room-temperature condition, it was seen on the third day in the low-temperature counterpart. Similarly, our Western blotting analysis showed high protein expression of CASP9 in the ovarian tissues stored at room temperature, especially on the first day of storage. Caspase 9 is known as an initiator of apoptotic signals, the same as caspase 2, 8, 10 [25]. Our result indicated that, while follicle viability decreased as storage time increased at both temperatures, intensive signalling of apoptosis and degeneration of cells in ovarian tissues was already started on the first day in the ovaries stored at room temperature. On the other hand, maintenance of a low expression level of *CASP9* until the second day of storage in the low-temperature condition suggested that this condition minimized the effect of storage time on cell death of ovarian tissues. The superiority of low-temperature storage was also confirmed at the protein level: In ovaries stored at a low temperature, CVH expression was maintained, while CASP9 expression was kept low through the 3 days of storage. These results demonstrated that keeping the ovaries at a low temperature makes them less prone to apoptosis and maintains their function as germ cells. In contrast, it was shown that storage conditions did not change the expression of *CASP3*. This is unexpected because caspase 9 is known to directly cleave and activate caspase 3 [25]. On the other hand, caspase 3 also has other activators, including caspase 8 and 10 as well as caspase 12, each of which is induced separately and has distinct roles in apoptosis. For instance, while caspase 12 is induced by endoplasmic reticulum (ER) stress, caspase 8 is activated by dimerization and self-cleavage at the death-inducing signalling complex with caspase 10, and both activate apoptosis through caspase 3 [25]. On the other hand, caspase 9 is a mitochondrial apoptotic mediator and is induced by a mitochondrion-centred cell death that is mediated by mitochondrial outer membrane permeabilization (MOMP), resulting in increased production of reactive oxygen species (ROS) following cytochrome c release [25,26]. The generation of ROS can provoke damage to multiple cellular organelles and processes, which can ultimately disrupt normal physiology [27]. In a pig study, ovary storage for longer than 2 h at 25 °C and 35 °C resulted in the low pH of the follicular fluid and high ROS level in immature oocytes [24]. In the present study, decreases in pH in the storage medium occurred prior to the increase in *CASP9* and matched the decrease in follicular integrity. Moreover, the expression of *SOD1*, the oxidation stress marker, tended to be higher in the room-temperature group than the low-temperature counterpart. Therefore, it is possible that production of ROS was induced by oxidation stress through the activation of caspase 9, resulting in the death of follicles in stored ovarian tissues. Because it is possible that oxidation stress is one of the main factors causing the degeneration of follicles during ovary storage, it would be worthwhile to investigate the changes in ROS within the stored ovaries and also the effect of antioxidants in the storage medium on follicle survival in the future.

It is worth noting that the influence of different storage temperatures was minimal in terms of follicle viability and morphology on the first day of ovary storage. This result was attractive, especially for conservation purposes of endangered avians. For example, in Okinawa rail, the body is not found immediately after death in their habitat and is left at room temperature until found, and then, in most cases, the ovaries are not recovered until at least 1 day after death. Although the environment surrounding the ovaries varies greatly depending on whether they are inside or outside of the body, and our gene/protein expression analysis and changes in the pH of the storage medium showed that the negative influence of room temperature has already started 1 day after storage as well as in the environment, it is possible that there is still a chance to recover the surviving follicle within the ovaries of birds left at room temperature for 1 day.

The vitrification study demonstrated that our storage protocol at low temperatures is suitable for maintaining the follicle integrity, including cryotolerance. To our knowledge, this is the first study to evaluate the effect of the storage period on follicle integrity after vitrification in avian ovaries. Since oocyte physiology is highly species-specific and commonly used cryoprotectants can also cause cell damage due to toxicity [28,29], choosing the proper agents and concentrations to reduce their toxicity is key in the development of a successful vitrification protocol of ovarian tissue. In this study, the use of KAv-1, specifically adjusted for avian cells with chick serum [20], resulted in a similar morphology normally compared to that of hen ovarian tissues in each storage period, suggesting that our freezing protocol could be effective for avian ovarian tissue vitrification. Our result is consistent with other vitrification studies in hen ovary [6], which successfully maintained follicle morphology with similar vitrification solutions except for the presence of chick serum in our study. The same group also reported the further improvement in follicle viability of hen ovarian tissues by supplementation of dietary polyunsaturated fatty acid and vitamin E as additional cryoprotectants [7]. A study in Japanese quail by another group showed that cell viability (but not follicle viability) in vitrified ovarian tissues were evaluated by trypan blue [8]. In our current study, while viability of ovarian cells was decreased after ovarian tissue vitrification, the authors successfully produced offspring from cryopreserved ovarian tissues by transplantation [8]. The difference between these previous studies and our study is that our study evaluated the ovarian tissue vitrification by not only histological observation or ovarian cell survival, but also by follicle viability using the NR staining assay. Although NR staining in the present study demonstrated that the ovarian tissues vitrified after ovary storage decreased the follicle viability, it also revealed the maintenance of follicle survival in vitrified ovarian tissues even after 3 days of ovary storage, suggesting the possibility of application for future transplantation to generate offspring. To increase follicle viability in vitrified hen ovaries, further improvement of the vitrification protocol is essential for applications of avian female fertility preservation. Nevertheless, our storage and vitrification protocol can serve as a useful shipping and cryopreservation method for avian ovaries to preserve follicle integrity, and future improvements to this system will be made.

This is the first study to evaluate the viability of avian ovarian follicles by NR staining, and we also validated the value of performing NR staining to confirm follicular survival in ovarian tissues. Although trends of decrease in follicular integrity by increase in the storage period were the same between before and after vitrification in hen ovarian tissues, the difference between before and after vitrification was significant throughout the storage period when analysed by NR staining. This is unexpected because the mean percentage of follicle integrity evaluated by NR staining and histology was similar in the ovarian tissues before vitrification. It could be suggested that changes in follicular integrity cannot be detected based on histological analysis. This result is consistent with our previous study in vitrified canine ovaries. In this study, the xenotransplantation assay revealed the importance of evaluating the long-term follicle integrity by evidence of a loss of follicle integrity in the frozen tissues considered to have morphologically normal follicles by histological observation [30]. In avian species, at least to our knowledge, there are no studies about in vitro culture nor xenotransplantation of ovarian tissues, and it is difficult to assess the long-term survival of ovarian follicles except for orthotopic ovarian transplantation. Therefore, NR staining provides a useful assay for evaluating the follicle integrity in avian ovarian tissues.

## 5. Conclusions

In conclusion, immature follicles in the stored chicken ovaries can survive for 3 days at 4 °C, but not at room temperature. The vitrification study proved that some follicles within chicken ovarian tissues can be cryopreserved even after 3 days of storage at 4 °C. Our study verifies the fact that the storage of hen ovaries for long periods of time and at low temperatures protects follicles that can be preserved by vitrification, but it also contributes to elucidating the effect of ovary storage on the integrity and cryotolerance of immature follicles within the hen ovary. Our study could provide a promising shipping protocol for ovaries for female gamete preservation in wild endangered avian species.

## Figures and Tables

**Figure 1 animals-12-01434-f001:**
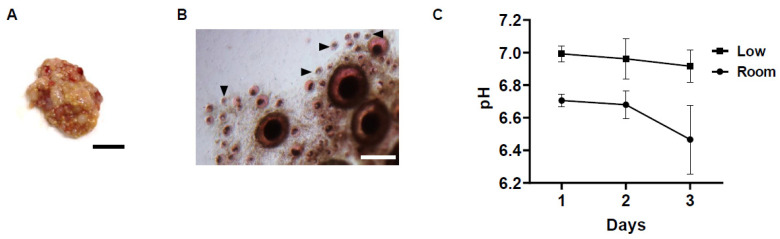
Photograph of average hen ovary utilized for this study, ovarian tissues stained with NR and changes in the pH value of the ovary storage medium. (**A**) Photograph of hen ovary for examination. Bar = 1 cm. (**B**) Representative photomicrograph of chicken ovarian tissue (stored for 2 days at low temperature) stained with NR after enzyme treatment. Most follicles were stained with NR, considered as viable. Arrowheads depict uncoloured follicles, considered as dead. Bar = 1 mm. (**C**) Mean pH (±SEM) changes of media in which the ovaries were stored under different conditions (low temperature, room temperature for 1 to 3 days).

**Figure 2 animals-12-01434-f002:**
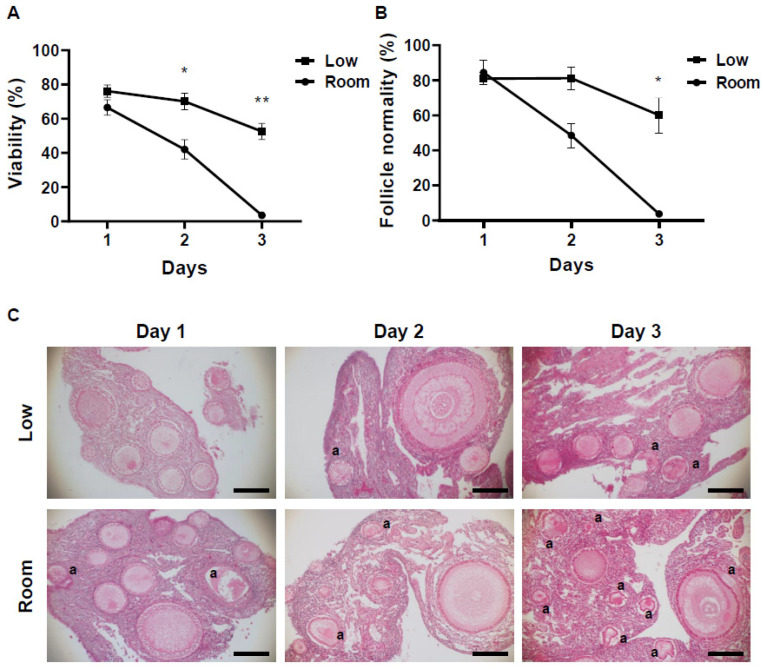
Influence of different storage conditions on follicle viability within ovarian tissues evaluated by neutral red (NR) and histological analysis. (**A**) Mean (± SEM) percentages of follicle viability in different conditions (low temperature, room temperature for 1 to 3 days). * indicate statistically significant differences (* *p* < 0.005, ** *p* < 0.0001) in the percentages of survived follicles. (**B**) Mean (± SEM) percentages of morphologically normal follicle in hen ovary stored in different conditions (low temperature, room temperature for 1 to 3 days). * indicate statistically significant differences (*p* < 0.05) in the percentages of morphologically normal follicles. (**C**) Histomicrographs of ovarian tissue stored in different conditions (low temperature, room temperature for 1 to 3 days). “a” indicates the abnormal follicles. Bar = 100 µm.

**Figure 3 animals-12-01434-f003:**
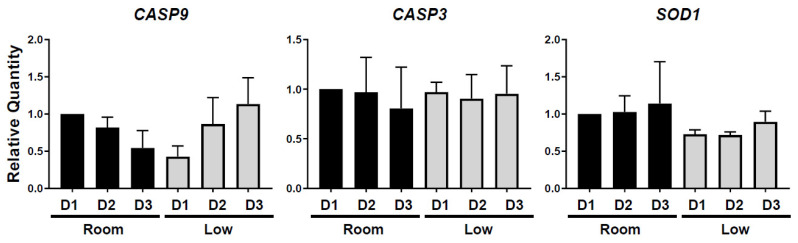
Influence of different storage condition on *CASP3*, *CASP9* and *SOD1* mRNA expressions in hen ovarian tissue. Mean (±SEM) percentages of relative gene expression in different conditions (low temperature, room temperature for 1 to 3 days).

**Figure 4 animals-12-01434-f004:**
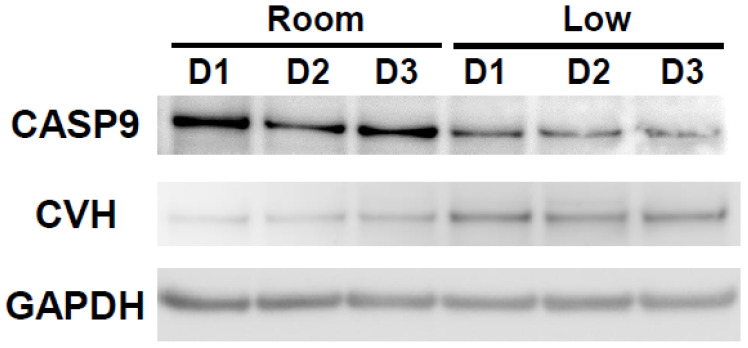
CASP9 and CVH protein expressions in hen ovarian tissues stored in different conditions. Representative blot for CASP9, CVH and GAPDH determined by western blot analysis.

**Figure 5 animals-12-01434-f005:**
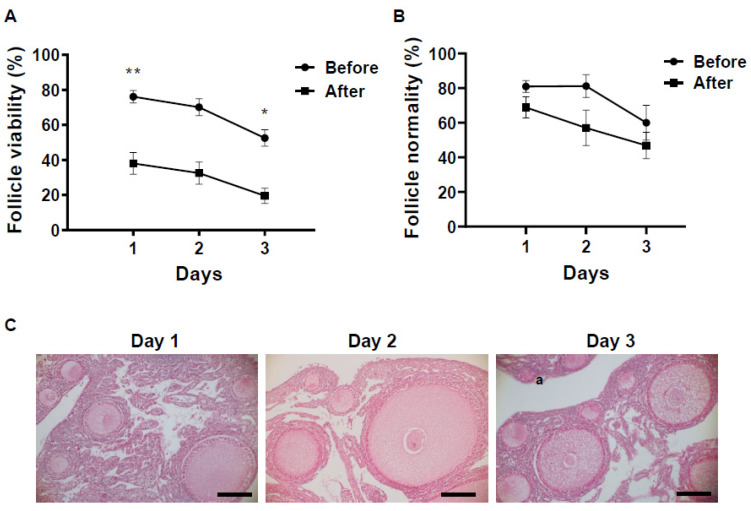
Influence of storage periods on follicle viability after vitrification of ovarian tissues (**A**) Mean (± SEM) percentages of follicle viability after vitrification of ovarian tissues stored for different periods (for 1 to 3 days). * indicate statistically significant differences (* *p* < 0.005, ** *p* < 0.001) in the percentages of survived follicles. (**B**) Mean (± SEM) percentages of morphologically normal follicle in ovarian tissues vitrified after ovary storage for different periods (for 1 to 3 days). (**C**) Histomicrographs of ovarian tissues vitrified after ovary storage for different periods stored at low temperature for 1 to 3 days). “a” indicates the abnormal follicles. Bar = 100 µm.

**Table 1 animals-12-01434-t001:** Primers for quantitative reverse transcription PCR (qPCR).

Primers		Sequence	Accession No.
Chicken CASP3	Forward	AGCAGGGAAACCCAAACTCT	NM_204725.1
	Reverse	CTGGTCCACTGTCTGCTTCA	
Chicken CASP9	Forward	CCCACTCCTGGTGACATCTT	XM_046903261.1
	Reverse	AGGTTTCCACGTACCACGAG	
Chicken SOD1	Forward	GTGGTCCATGCAAAAAGTGA	NM_205064.1
	Reverse	AAGCTAAACGAGGTCCAGCA	
Chicken GAPDH	Forward	TGGGAAGCTTACTGGAATGG	NM_204305.1
	Reverse	CTTGGCTGGTTTCTCCAGAC	

## Data Availability

The datasets generated and/or analysed during this study are available from the corresponding author upon reasonable request.

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
