# Peer review of "Cryopreservation Competence of Chicken Oocytes as a Model of Endangered Wild Birds: Effects of Storage Time and Temperature on the Ovarian Follicle Survival"

_animals, 2022, doi:10.3390/ani12111434_

Round 1

Reviewer 1 Report

1, In “Figure 2.(C) Histomicrographs of ovarian tissue stored in different conditions (low temperature, room temperature for 1 to 3 days)”. The histomicrograph of day2/room should be replaced by one with some morphologically abnormal follicles.

2, In “Figure 5.(C) Histomicrographs of ovarian tissues vitrified after ovary storage for different periods stored at low temperature for 1 to 3 days). The histomicrograph of day2 should be replaced by one with some morphologically normal follicles compared to picture of day3. The arrangement of sequence or the selection of your pictures make readers feel puzzled.

3, if western blot analysis of CASP3 and SOD1 provided, the manuscript will become much better.

Author Response

Dear Reviewer #1,

First of all, we very much appreciate your constructive comments in our manuscript. To address the comments, we have revised the manuscript as follows:

Reviewer’s comment 1;

In “Figure 2.(C) Histomicrographs of ovarian tissue stored in different conditions (low temperature, room temperature for 1 to 3 days)”. The histomicrograph of day2/room should be replaced by one with some morphologically abnormal follicles.

Our response;

We followed the reviewer’s suggestion and replaced the picture of “Day 2/room” in Figure 2C.

Reviewer’s comment 2;

In “Figure 5.(C) Histomicrographs of ovarian tissues vitrified after ovary storage for different periods stored at low temperature for 1 to 3 days). The histomicrograph of day2 should be replaced by one with some morphologically normal follicles compared to picture of day3. The arrangement of sequence or the selection of your pictures make readers feel puzzled.

Our response;

We followed the reviewer’s suggestion and replaced the picture of “Day 2/room” in Figure 5C.

Reviewer’s comment 3;

If western blot analysis of CASP3 and SOD1 provided, the manuscript will become much better.

Our response;

We agree with the reviewer’s comments that it is better to analyze CASP3 and SOD1 by western blotting. Actually, we had done western blotting analysis with the CASP3 antibody, but the band could not be confirmed at the expected size, and its expression could not be confirmed. This is probably due to the difficulty to select the antibody which works for birds because most of the antibody were made for experimental animals in mammals, even though the product information says it works for birds. Nevertheless, we think that the results of CASP9 and CVH protein expression support our finding that ovaries stored under low temperature conditions maintained germ cells while suppressing apoptosis.

Reviewer 2 Report

The manuscript presents incredibly important issues which should be taken account regarding to wild birds gametes preservation.

Although the manuscript brings interesting information, it contains several issues that need to be addressed.

  • Introduction: Please check carefully again the article cited [17], because their conclusions are contrary to what you wrote
  • Materials and methods: please check if tissue can be stored at -80C in RNA Later?
  • Division of samples number is not clear. Why 30 hens were used if you then used just 3 (n) for some experiment? There is written: “At least three pieces of ovarian tissues from each donor under various storage conditions  were utilized for evaluation of storage condition and the rest of the ovarian tissues stored at 4 °C were vitrified after different storage time.” Then it is written n=4 were examined with NR staining. Moreover, it is not corrected to perform experiment eg RTPCR on 3 samples (even with 3 replication). How statistics could be calculated then?
  • How many samples were checked by Western Blott?
  • Why didn’t you check gene expression after tissue cryopreservation?
  • Only histological evaluation is not relevant, unfortunately. My personal experience allows me to stated that even you received structurally intact follicles – there was apoptosis.
  • Why in RTPCR the sample stored at room temperature for 1 day was the control?! In my opinion it is methodological mistake – it should be fresh sample just after ovary collection.

In my opinion, because of lack of gene expression results after vitrification this study appear to be incomplete.

I did not make any suggestions concerning the language because it would be too time consuming and not done as well as it would be by professionals. I suggest the Authors to send the paper to an appropriate company.

Author Response

Dear Reviewer #2,

First of all, we appreciate very much for the reviewer’s positive tone regarding our study and constructive comments in our manuscript.

To address the comments, we have revised the manuscript as follows:

Reviewer’s comment 1;

The manuscript presents incredibly important issues which should be taken account regarding to wild birds gametes preservation. Although the manuscript brings interesting information, it contains several issues that need to be addressed.

Introduction: Please check carefully again the article cited [17], because their conclusions are contrary to what you wrote

Our response;

We verified the citation carefully and we agree that storage temperature affected the meiotic competence in canine ovaries in this article. We revised this part of current manuscript (Line 96) and adjusted the reference numbers by this change (from [17] to [16]).

Reviewer’s comment 2;

Materials and methods: please check if tissue can be stored at -80C in RNA Later?

Our response;

We confirmed with the company that the tissues can be stored at -80°C in RNA Later.

Reviewer’s comment 3;

Division of samples number is not clear. Why 30 hens were used if you then used just 3 (n) for some experiment? There is written: “At least three pieces of ovarian tissues from each donor under various storage conditions were utilized for evaluation of storage condition and the rest of the ovarian tissues stored at 4 °C were vitrified after different storage time.” Then it is written n=4 were examined with NR staining.

Moreover, it is not corrected to perform experiment eg RTPCR on 3 samples (even with 3 replication). How statistics could be calculated then?

Our response;

I apologize for the confusion. We believe that the advantage of our study is the use of chickens as a model animal of endangered birds. However, because chickens were kept in a different facility, this led to the limitation that we could not obtain the chicken ovaries in a routinely fashion and thus they were collected randomly. Each chicken ovary was preserved under one storage condition to examine the influence of storage conditions. After recovery from storage medium, ovaries were cut into small pieces and then divided into different experiments as well as vitrification. Because we could not process the experiments as we planned every time, the number of animals and the number of replicates did not match in some experiments/conditions. Also, because follicles were not uniformly distributed within the ovaries, we used 2 to 3 pieces/hens/condition in each ovary per experiments.

In detail, in study 1, we examined 5 replicates of 6 different storage conditions (3 storage period (Day 1, 2 and 3) of room vs. low temperature), resulting in a total of 30 hens. All of 5 replicates were used for pH measurements (n = 5 per condition) and 4 of them were used for viability analysis with neutral red staining as well as histological analysis (n = 4 per condition in each analysis). Then, 3 of them were used for RNA analysis and western blotting analysis (n = 3 per condition in each analysis).

In study 2, we used 4 of same 5 replicates (3 storage period in low temperature condition) of study 1 as we described above. All of 4 replicates were used for viability analysis with neutral red staining and histological analysis (n = 4 per condition in each analysis). Then, 3 of them were used for additional RNA analysis as we explained for reviewers as follows (Supplementary Figure for reviewer).

For the concern regarding the number of replicates in real time RT-PCR, the sufficient number that can be tested with ANOVA was carried out and data of each sample were obtained from three technical replicates (triplicates) in the experimental set-up of a real time RT-PCR reaction.

To avoid confusion, we revised the methods parts (Line 130-133, 224-240).

Reviewer’s comment 4;

How many samples were checked by Western Blott?

Our response;

We examined with three replicates as written in the materials and methods part (Line 231-232) and confirmed that all three showed similar band patterns.

Reviewer’s comment 5;

Why didn’t you check gene expression after tissue cryopreservation?

Only histological evaluation is not relevant, unfortunately. My personal experience allows me to stated that even you received structurally intact follicles – there was apoptosis.

In my opinion, because of lack of gene expression results after vitrification this study appear to be incomplete.

Our response;

We agree with the reviewer’s comments that only histological evaluation is not enough to analyze the viability of ovarian follicles as we also mentioned at discussion part (Line 436-445). Therefore, in addition to histology, we performed neutral red staining in vitrified ovarian tissues in the submitted manuscript and we believe this makes our finding stronger and also unique as mentioned in a discussion part (Line 434-436, 445-449).

Further, as followed by the reviewer’s suggestion, we conducted an additional experiment of real time qPCR in vitrified ovarian tissues to compare the influence of ovary storage period (Supplementary figure for reviewers, n = 3). As a results, expression levels of SOD1 and CASP9 did not change among storage period in vitrified tissues. In addition, expression level of CVH (Chicken VASA Homologue) in vitrified tissues did not change among storage period of ovary, which indicates that preserved germ cells in vitrified tissues did not change during the 3 days storage. Only CASP3 expressions was affected by ovary storage period as expression level decreased in vitrified tissues after 3 days storage compared to 1 day storage.

However, RNA expression is affected by subtle changes in conditions of cryopreservation each time and we think it needs more scrutiny before it is included in the current paper. Moreover, although it may require additional study to investigate the maintenance of detailed functionality in vitrified ovarian tissues, the follicle survival in vitrified ovarian tissues after ovary storage at 4 °C is one of the main findings in our manuscript. We believe the verification by the two types of experiments (neutral red staining and histology) supports our research result in a sufficient manner. Therefore, we would like not to include this additional experiment/figure in the revised manuscript.

Reviewer’s comment 6;

Why in RTPCR the sample stored at room temperature for 1 day was the control?! In my opinion it is methodological mistake – it should be fresh sample just after ovary collection.

Our response;

We understand your concern. However, we conducted this study as a model of gamete preservation in wild birds and it is normal that ovary is left within the body at room temperature and field site where the ovary is recovered is far from the laboratory facilities as mentioned in an introduction part (Line 87-92). Similarly, the animal facility of our current experiments is far from the laboratory performing the experiments. That’s why we didn’t put the fresh control and we used samples stored at room temperature condition for 1 day as a calibrator to compare all the samples stored in different conditions.

Reviewer’s comment 7;

I did not make any suggestions concerning the language because it would be too time consuming and not done as well as it would be by professionals. I suggest the Authors to send the paper to an appropriate company.

Our response;

Thank you for your kind suggestion. We requested an English editing service, MDPI to improve the English and revised our whole manuscript.

Round 2

Reviewer 2 Report

As already mentioned, the manuscript presents extremely important issues that must be considered in the protection of the gametes of wild birds.

However, the study was designed with huge flaw. This is the number of animals on which each condition has been checked. In my opinion, performing RTPCR and Western blots on 3-4 animals is unacceptable in research studies.

I strongly encourage the authors to return with such research in which they will have at least 6 animals tested in the group